# Universal Approximation Under Constraints is Possible with Transformers

**Anastasis Kratsios,**[*] **Tianlin Liu & Ivan Dokmanić**
Universität Basel,
Departement Mathematik und Informatik
{firstname.lastname}@unibas.ch

**Behnoosh Zamanlooy**[*]
Universität Zürich,
Department of Informatics
bzamanlooy@ifi.uzh.ch

## Abstract

Many practical problems need the output of a machine learning model to satisfy a set of constraints, $K$. There are, however, no known guarantees that classical neural networks can exactly encode constraints while simultaneously achieving universality. We provide a quantitative constrained universal approximation theorem which guarantees that for any convex or non-convex compact set $K$ and any continuous function $f : \mathbb{R}^n \to K$, there is a probabilistic transformer $\hat{F}$ whose randomized outputs all lie in $K$ and whose expected output uniformly approximates $f$. Our second main result is a "deep neural version" of Berge (1963)'s Maximum Theorem. The result guarantees that given an objective function $L$, a constraint set $K$, and a family of soft constraint sets, there is a probabilistic transformer $\hat{F}$ that approximately minimizes $L$ and whose outputs belong to $K$; moreover, $\hat{F}$ approximately satisfies the soft constraints. Our results imply the first universal approximation theorem for classical transformers with exact convex constraint satisfaction, and a chart-free universal approximation theorem for Riemannian manifold-valued functions subject to geodesically-convex constraints.

**Keywords**: Constrained Universal Approximation, Probabilistic Attention, Transformer Networks, Geometric Deep Learning, Measurable Maximum Theorem, Non-Affine Random Projections.

## 1 Introduction

In supervised learning, we select a parameterized model $\hat{f} : \mathbb{R}^n \to \mathbb{R}^m$ by optimizing a real-valued loss function[1] $L$ over training data from an input-output domain $\mathcal{X} \times \mathcal{Y} \subseteq \mathbb{R}^n \times \mathbb{R}^m$. A necessary property for a model class to produce asymptotically optimal results, for any continuous loss $L$, is the universal approximation property. However, often more structure (beyond vectorial $\mathbb{R}^m$) is present in a learning problem and this structure must be encoded into the trained model $\hat{f}$ to obtain meaningful or feasible predictions. This additional structure is typically described by a constraint set $K \subseteq \mathbb{R}^m$ and the condition $\hat{f}(\mathcal{X}) \subseteq K$. For example, in classification $K = \{y \in [0,1]^m : \sum_{i=1}^m y_i = 1\}$ (Shalev-Shwartz & Ben-David, 2014), in Stackelberg games (Holters et al., 2018; Jin et al., 2020; Li et al., 2021) $K$ is the set of utility-maximizing actions of an opponent, in integer programming $K$ is the integer lattice $\mathbb{Z}^m$ (Conforti et al., 2014), in financial risk-management $K$ is a set of positions meeting the minimum solvency requirements imposed by international regularity bodies (Basel Committee on Banking Supervision, 2015; 2019; McNeil et al., 2015), in covariance matrix prediction $K \subseteq \mathbb{R}^{m \times m}$ is the set of $m \times m$ matrices which are symmetric and positive semi-definite (Bonnabel et al., 2013; Bonnabel & Sepulchre, 2009; Baes et al., 2021), in geometric deep learning $K$ is typically a manifold (e.g. a pose manifold in computer vision and robotics (Ding & Fan, 2014) or a manifold of distance matrices (Dokmanic et al., 2015)), a graph, or an orbit of a group action (Bronstein et al., 2017; 2021; Kratsios & Bilokopytov, 2020). Therefore, we ask:

*Is exact constraint satisfaction possible with universal deep learning models?*

---

[*]Corresponding authors.

[1]For example, in a regression problem one can set $L(x, y) = \|f(x) - y\|$ for an unknown function $f$ or in regression problems one sets $L(x, y) = \sum_{i=1}^m [C(x)]_i \log(y_i)$ for an unknown classifier $C$.

The answer to this question begins by examining the classical universal approximation theorems for deep feedforward networks. If $L$ and $K$ are mildly regular, the universal approximation theorems of Hornik et al. (1989); Cybenko (1989); Pinkus (1999); Gühring et al. (2020); Kidger & Lyons (2020); Park et al. (2021) guarantee that for any "good activation function $\sigma$" and for every tolerance level $\epsilon > 0$, there is a deep feedforward network with activation function $\sigma$, such that $\inf_{y \in K} L(x, y)$ and $L(x, \hat{f}(x))$ are uniformly at most $\epsilon$ apart. Written in terms of the optimality set,

$$\sup_{x \in \mathcal{X}} \| \hat{f}(x) - \operatorname*{argmin}_{y \in K} L(x,y) \| \leq \epsilon, \tag{1}$$

where the distance of a point $y \in \mathbb{R}^m$ to a set $A \subseteq \mathbb{R}^m$ is defined by $\|y - A\| \triangleq \inf_{a \in A} \|y - a\|$. Since $\operatorname{argmin}_{y \in K} L(x, y) \subseteq K$, then (1) only implies that $\|\hat{f}(x) - K\| \leq \epsilon$ and there is no reason to believe that the constraint $\hat{f}(x) \in K$ is exactly satisfied, for every $x \in \mathcal{X}$.

This kind of *approximate* constraint satisfaction is not always appropriate. In the following examples constraint violation causes either practical or theoretical concerns:

(i) In post-financial crisis risk management, international regulatory bodies mandate that any financial actor should maintain solubility proportional to the risk of their investments (Basel Committee on Banking Supervision, 2015; 2019). To prevent future financial crises, any violation of these risk constraints, no matter the size, incurs large and immediate fines.

(ii) In geometric deep learning, we often need to encode complicated non-vectorial structure present in a dataset, by viewing it as a $K$ valued function (Fletcher, 2013; Bonnabel & Sepulchre, 2009; Baes et al., 2021). However, if $K$ is non-convex then Motzkin (1935) confirms that there is no unique way to map predictions $\hat{f}(x) \notin K$ to a closest point in $K$. Thus, we are faced with the dilemma: either make an ad-hoc *choice* of a $k$ in $K$ with $k \approx \hat{f}(x)$ (ex.: an arbitrary choice scheme when $K = \mathbb{Z}^m$) or have meaningless predictions (ex: non-integer values to integer programs, or symmetry breaking (Weinberg, 1976)[2]).

Constrained learning was recognized as an effective framework for fairness and robustness by Chamon & Ribeiro (2020) who study empirical risk minimization under constraints. Many emerging topics in machine learning lead to constrained learning formulations. A case in point is model-based domain generalization (Robey et al., 2021). Despite the importance of (deep) learning with constraints, there are no related approximation-theoretic results to the best of our knowledge.

In this paper, we bridge this theoretical gap by showing that universal approximation with exact constraint satisfaction is always possible for deep *(probabilistic) transformer networks* with a single attention mechanism as output layer. Our contribution is three-fold:

1. We derive the first universal approximation theorem with exact constraint satisfaction;
2. Our transformer network's encoder and decoder adapt to the dimension of the constraint set and thus beat the curse of dimensionality for low-dimensional constraints;
3. Our models leverage a probabilistic attention mechanism that can encode non-convex constraints. This probabilistic approach is key to bypass the topological obstructions to non-Euclidean universal approximation (Kratsios & Papon, 2021).

Our analysis provides perspective on the empirical success of attention and adds to the recent line of work on approximation theory for transformer networks, (Yun et al., 2020a;b), which roughly considers the unconstrained case (with $K$ in (1) replaced by $\mathbb{R}^m$) in the special case of $L(x, y) = \|f(x) - y\|$ for a suitable target function $f : \mathbb{R}^n \to \mathbb{R}^m$. Our probabilistic perspective on transformer networks fits with the representations of Vuckovic et al. (2021) and of Kratsios (2021).

Our results can be regarded as an approximation-theoretic counterpart to the constrained statistical learning theory of Chamon & Ribeiro (2020). Further, they put forward a perspective on randomness in neural networks that is complementary to the work of Louart et al. (2018); Gonon et al. (2020a;b). We look at the same problem focusing on constraint satisfaction instead of training efficiency. Finally, our proof methods are novel, and build on contemporary tools from metric geometry (Ambrosio & Puglisi, 2020; Bruè et al., 2021).

---

[2]As discussed in Rosset et al. (2021) this is problematic since respecting symmetries can often massively reduce the computational burden of a learning task.

## 1.1 THE PROBABILISTIC ATTENTION MECHANISM

We now give a high-level explanation of our results; the detailed formulations are in Section 2.

Introduced in (Bahdanau et al., 2015) and later used to define the transformer architecture (Vaswani et al., 2017), in the NLP context, *attention* maps a matrix of queries $Q$, a matrix of keys $K$, and a matrix of values $V$ to the quantity $\mathrm{Softmax}(QK^\top)V$, where the softmax function (defined below) is applied row-wise to $QK^\top$. Just as the authors of (Petersen & Voigtlaender, 2020; Zhou, 2020) focus on the simplified versions of practically implementable ConvNets in the study of approximation theory of deep ConvNets (e.g. omitting pooling layers), we find it sufficient to study the following simplified attention mechanism to obtain universal approximation results:

$$\mathrm{Attention}\,(w, Y) \triangleq \mathrm{Softmax}_N\,(w)^\top Y = \sum_{n=1}^N [\mathrm{Softmax}_N(w)_n] Y_n, \tag{2}$$

where $w \in \mathbb{R}^N$, $\mathrm{Softmax}_N : \mathbb{R}^N \ni w \mapsto (\frac{e^{w_k}}{\sum_{j=1}^N e^{w_j}})_{k=1}^N$, and $Y$ is an $N \times m$ matrix. The attention mechanism (2) can be interpreted as "paying attention" to a set of particles $Y_1, \dots, Y_N \in \mathbb{R}^m$ defined by $Y$'s rows. This simplified form of attention is sufficient to demonstrate that transformer networks can approximate a function while respecting a constraint set, $K$, whether convex or non-convex.

**Informal Theorem 1.1** (Deep Maximum Theorem for Transformers). *If $K$ is convex and the quantities defining (1) are regular then, for any $\epsilon \in (0, 1]$, there is a feedforward network $\hat{f}$, an $\mathcal{X}_\epsilon \subset \mathbb{R}^n$ of probability 1-$\epsilon$, and a matrix $Y$ such that the transformer $\mathrm{Attention}(\hat{f}(x), Y)$ satisfies:*

*(i) **Exact Constraint Satisfaction:** For each $x \in \mathbb{R}^n$, $\mathrm{Attention}(\hat{f}(x), Y) \in K$,*

*(ii) **Universal Approximation:** $\sup_{x \in \mathcal{X}_\epsilon} \| \mathrm{Attention}(\hat{f}(x), Y) - \underset{y^\star \in K}{\mathrm{argmin}} L(x, y^\star)\| \leq \epsilon$*

Informal Theorem 1.1 guarantees that simple transformer networks can minimize any loss function while exactly satisfying the set of convex constraints. As illustrated by Figure 1 and Figure 2, $K$'s convexity is critical here, since without it the transformer's prediction may fail to lie in $K$. This is because any transformer network's output is a convex combinations of the *particles* $Y_1, Y_2, Y_3$; thus, any transformer network's predictions must belong to these particles' convex hull.

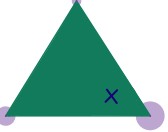

Figure 1: Convex Constraints

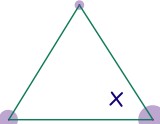

Figure 2: Non-Convex Constraints

In Figures 1 and 2, $Y$'s columns, i.e. the particles $Y_1, Y_2$, and $Y_3$, are each illustrated by a ● at the constraint set ($K$) vertices. The bubble around each each $Y_i$ illustrates the predicted probability, for a given input, that $f(x)$ is nearest to that $Y_i$. The × is the transformer's prediction which is, by construction, a convex combination of the $Y_i$ weighted by the aforementioned probabilities and therefore they lie in the $K$ if it is convex (Figure 1) but not if $K$ is non-convex (Figure 2).

Naturally, we arrive at the question: *How can (i) and (ii) simultaneously hold when $K$ is non-convex?*

Returning to Vaswani et al. (2017) and using the introduced terminology, we note that the role of the $\mathrm{Softmax}_N$ layer is to rank the importance of the particles $\{Y_n\}_{n=1}^N$ when optimizing $L$, at any given input: the weights $[\mathrm{Softmax}_N(w)]_n$ in (2) can be interpreted as charging their respective *point masses* $\{\delta_{Y_n}\}_{n=1}^N$ with probabilities of being optimal for $L$ (relative to the other particles)[3]. This suggests the following *probabilistic reinterpretation of attention* (which we denote by p-attention):

$$\mathrm{P\text{-}attention}(w, Y) \triangleq \sum_{n=1}^N [\mathrm{Softmax}_N(w)]_n \delta_{Y_n}. \tag{3}$$

---

[3]Following Villani (2009), $\delta_{Y_n}$ is the Borel probability measure on $\mathbb{R}^m$ assigning full probability to any Borel subset of $\mathbb{R}^m$ containing the particle $Y_n$ and 0 otherwise.

Crudely put, P-attention$(\cdot, Y)$ *"pays relative attention to the particles"* $Y_1, \ldots, Y_n \in \mathbb{R}^m$.

A simple computation shows that the mean prediction of our probabilistic attention mechanism, exactly implements "classical" Attention of Vaswani et al. (2017), as defined in (2),

$$\text{Attention}(w, Y) = \mathbb{E}_{X \sim \text{P-attention}(w,Y)}[X], \tag{4}$$

where $\mathbb{E}_{X \sim \text{P-attention}(w,Y)}[X]$ denotes the (vector-valued) expectation of a random-vector $X$ distributed according to P-attention$(w, Y)$. Hence, (3) is no less general than (2). The advantage of (3) is that, if each particle $Y_n$ belongs to $K$ (even if $K$ is non-convex) then, any sample drawn from the probability measure P-attention$(w, Y)$ necessarily belongs to $K$.

## 1.2 Qualitative Results: Deep Maximum Theorem

Probabilistic attention (3) yields the following non-convex generalization of Informal Theorem 1.1. The result is a *qualitative universal approximation theorem* as well as a deep neural version of the Maximum Theorem[4] (Berge, 1963), which states that under mild regularity conditions, given any well-behaved family of input dependent "soft constraint sets" $\{C_x\}_{x \in \mathbb{R}^n}$ compatible with $K$, there is a measurable function mapping each $x \in \mathbb{R}^n$ to a minimizer of $L(x, y)$ on $K \cap C_x$.

We use $\mathscr{W}_1$ to denote the Wasserstein-1 distance between probability measures on $K$. The results also give the flexibility to the user to enforce an input-dependent family of "soft constraints" $\{C_x\}_{x \in \mathbb{R}^n}$ which only need to hold approximately; definitions are provided in Section 1.4.

**Informal Theorem 1.2** (Deep Maximum Theorem: Non-Convex Case). *If the quantities defining 1 are regular, $K$ is a compact set of "exact constraints", and $\{C_x\}_{x \in \mathbb{R}^n}$ a set of "soft constraints", then, for any approximation quality $0 < \epsilon \leq 1$, there is a deep feedforward network $\hat{f}$ and a matrix $Y$ satisfying:*

 *(i) **Exact Constraint Satisfaction:** For each $x \in \mathbb{R}^n$, P-attention$(\hat{f}(x), Y)$ is supported in $K$,*
 *(ii) **Universal Approximation:** $\mathbb{P}(\mathscr{W}_1(\text{P-attention}(\hat{f}(x), Y), \underset{y^\star \in C_x \cap K}{\arg\min} L(x, y^\star)) \leq \epsilon) \geq 1 - \epsilon;$*

*where for a probability measure $\mathbb{P}$ on $\mathbb{R}^m$ and a $B \subseteq \mathbb{R}^m$ we define $\mathscr{W}_1(\mathbb{P}, B) \triangleq \inf_{b \in B} \mathscr{W}_1(\mathbb{P}, \delta_b)$.*

**Example 1.3** (Reduction to Classical Point-to-Set Distance). *In particular, when $\mathbb{P}$ is a point-mass $\mathbb{P} = \delta_y$ for some $y \in \mathbb{R}^m$, then one recovers the familiar Euclidean distance to the set $B$ via:*

$$\mathscr{W}_1(\delta_y, B) \overset{(\text{def})}{=} \inf_{b \in B} \mathscr{W}_1(\delta_y, \delta_b) = \inf_{b \in B} \|y - b\| \overset{(\text{def})}{=} \|y - B\|;$$

*where the first and second equality follows from (Villani, 2009, (5) - page 99), and the last equality is the definition of $\|y - B\|$ (as in (Aubin & Frankowska, 2009, Definition 1.1.1)).*

Another important class of non-convex constraints arising from geometric deep learning where $K$ is a non-Euclidean ball in a Riemannian submanifold of $\mathbb{R}^m$. In this broad case, we may extract mean predictions from P-attention$(\hat{f}, Y)$, by applying the *Fréchet mean* introduced in Fréchet (1948). Such "geometric means" are well-understood theoretically (Bhattacharya & Patrangenaru, 2003) and easily handled numerically Miolane et al. (2020); Lou et al. (2020).

## 1.3 Quantitative Results: Constrained Universal Approximation Theorem

In its current form, the objective function $L$ is too general to derive quantitative approximation rates[5]. Nevertheless, as with most universal approximation theorems (Hornik et al., 1989; Pinkus, 1999; Kidger & Lyons, 2020), if each soft constraint $C_x$ is set to $\mathbb{R}^m$ and $L$ quantifies the uniform distance to an *unknown continuous function* $f : \mathbb{R}^n \to K$ in the Euclidean sense,

$$L(x, y) \triangleq \|f(x) - y\|,$$

then, Informal Theorem 1.2 reduces to a (qualitative) universal approximation for transformer networks with exact constraint satisfaction. In fact, this additional structure is enough for us to derive quantitative versions of the aforementioned results. We permit ourselves the general situation, where

---

[4]More precisely, our result is a deep neural version of the measure-theoretic counterpart to Berge's Maximum Theorem; see (Aliprantis & Border, 2006, (Measurable Maximum Theorem) - Theorem 18.19).

[5]For instance, $L$ can describe anything from a regression, to a clustering problem.

$K$ is contained in an unknown $d$-*dimensional* submanifold (where $d \in \Theta(m^{\frac{1}{s}})$ for some $s > 0$). Our approximation rates scale favourably in the ratio $s \approx \frac{\log(m)}{\log(d)}$; i.e., we avoid the curse of dimensionality for low-dimensional constraint sets. This additional structure translates into the familiar encoder-decoder structure deployed in most transformer network implementations.

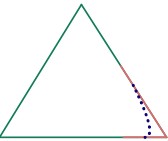

Figure 3: Encoder : $\approx f$

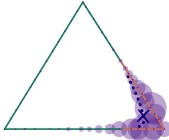

Figure 4: Decoder : $\approx$ Random Projection to $K$

Figure 3 illustrates the *encoder network* $\hat{\mathcal{E}} : \mathbb{R}^n \to \mathbb{R}^m$, whose role is to perform a (classical) unconstrained approximation of the target function, $f$. Since $\hat{\mathcal{E}}$ is a classical feedforward network then its approximation of the target function can be arbitrarily close to the constraint set $K$ but it need not lie in it. The next step is to "map the *encoder network*'s output onto $K$ with low distortion." The role of the *decoder network* $\hat{\mathcal{D}}$ is to correct any constraint violation made by *encoder network* by "projecting them back on to $K$". However, such a projection does not exist if $K$ is not convex since there must be more than one closest point in $K$ to some $y \in \mathbb{R}^m$ (Motzkin, 1935). Nevertheless, if the "projection" were capable of mapping any $y \in \mathbb{R}^m$ to *multiple* points on $K$, ranked by their proximity to $y$, then there would be no trouble. The *decoder network* accomplishes precisely this, as illustrated in Figure 4, where the bubbles illustrate the probability of any particle in $K$ being closest to $y$, illustrated by the size of the bubbles in Figure 4. Mathematically,[6] $\hat{\mathcal{D}} : \mathbb{R}^m \to \mathcal{P}_1(K)$ approximates a *(non-affine) random projection*, in the sense of Ohta (2009); Ambrosio & Puglisi (2020); Bruè et al. (2021); i.e.: a 1-Lipschitz map $\Pi : \mathbb{R}^m \to \mathcal{P}_1(K)$ satisfying the *random projection property*: *for all* $y \in K$

$$\Pi_y = \delta_y.$$

Thus, $\Pi$'s random projection property means that it fixes any output already satisfying the constraint $K$, and its Lipschitz regularity implies that it is stable. Thus, sampling from $\Pi(y_1)$ is similar to sampling from $\Pi(y_2)$ whenever the points $y_1, y_2 \in \mathbb{R}^m$ are near to one another.

**Remark 1.4.** *Random projections are closely tied to the (random) partitions of unity of Lee & Naor (2005) (see (Ambrosio & Puglisi, 2020, Theorem 2.8)). These random projections generalize the random projections of Johnson & Lindenstrauss (1984), beyond the case where $K$ is affine.*

**Remark 1.5.** *The special case of random projections onto affine spaces has recently been used when constructing universal neural models (Cuchiero et al., 2021; Puthawala et al., 2020).*

We record the complexity of both the decoder and encoder networks constructed in our quantitative results in Table 1. Here $A, B, C, D \geq 0$ are constants independent of $\epsilon$ and $k$, where $k \in \mathbb{N}_+$ is the number of continuous derivatives which $f$ admits (when viewed as a function into $\mathbb{R}^m$). From

| Network | $\hat{\mathcal{E}}$ | $\hat{\mathcal{D}}$ |
|---|---|---|
| Depth | $\mathcal{O}(m^{\frac{1}{s}}(1 + \epsilon^{\frac{2n}{3(kn+1)} - \frac{2n}{kn+1}}))$ | $\mathcal{O}\left((N^{\frac{3}{2}}(A + 2\epsilon)(4 - \epsilon^{-1})^2)^{\frac{2m}{s}}\right)$ |
| Width | $m^{\frac{1}{s}}(4n + 10)$ | $m^{\frac{1}{s}} + N + 2$ |
| $N$ | - | $\mathcal{O}\left((\epsilon^{-1}A + B)^{\frac{m}{2}}\right)$ |
| $Q$ | - | $\mathcal{O}\left(\epsilon^{\frac{-m}{s}}\right)$ |

Table 1: Complexity of simple transformer network $\hat{f} = \hat{\mathcal{D}} \circ \hat{\mathcal{E}}$ approximating $f$.

Table 1, we see that if $m^{\frac{1}{s}} \ll m$ then, $s > 0$ is large; hence, $\epsilon^{\frac{m}{s}}$, $(1 - 4\epsilon^{-1})^{\frac{2m}{s}}$, and $N^{\frac{m}{s}}$ are small.

### 1.4 NOTATION AND BACKGROUND

**Optimal Transportd**   Given any non-empty subset $K \subseteq \mathbb{R}^m$, the set of all Borel probability measures $\mathbb{P}$ on $K$ with a finite mean; i.e.: $\mathbb{E}_{X \sim \mathbb{P}}[\|X\|] < \infty$, is denoted by $\mathcal{P}_1(K)$. *Wasserstein*

---

[6]$\mathcal{P}_1(K)$ denotes the Wasserstein space on $K$, and is defined formally below.

*distance* $\mathscr{W}_1$ is defined for any $\mathbb{P}, \mathbb{Q} \in \mathcal{P}_1(K)$ by the minimal energy needed to transport all mass from $\mathbb{P}$ to $\mathbb{Q}$. Following Villani (2009), $\mathscr{W}_1(\mathbb{P}, \mathbb{Q})$ is defined by:

$$\mathscr{W}_1(\mathbb{P}, \mathbb{Q}) \triangleq \inf_{\pi} \mathbb{E}_{(X_1, X_2) \sim \pi}[\|X_1 - X_2\|],$$

where the infimum is taken over all Borel probability measures $\pi$ on $K^2$ with marginals $\mathbb{P}$ and $\mathbb{Q}$. The metric space $(\mathcal{P}_1(K), \mathscr{W}_1)$ is named the *Wasserstein space over* $K$; we abbreviate it by $\mathcal{P}_1(K)$.

**Smooth Function Spaces**    The set of real-valued continuous functions on $\mathbb{R}^n$ is denoted by $C(\mathbb{R}^n)$. Let $k \in \mathbb{N}_+$ and $\mathcal{X} \subseteq [0,1]^n$ be non-empty. The set of functions $f : \mathcal{X} \to K$ for which there is a $k$-times continuously differentiable $\boldsymbol{f} : \mathbb{R}^n \to \mathbb{R}^m$ extending $f$; i.e.: $\boldsymbol{f}|_{\mathcal{X}} = f$, is denoted by $C_{tr}^k(\mathcal{X}, K)$. Our interest in $C_{tr}^k(\mathcal{X}, K)$ does not stem from the fact that it contains all smooth functions mapping $[0,1]^n$ to $K$, but rather that it allows us to speak about the uniform approximation of discontinuous $K$-valued functions on regions in $[0,1]^n$ where they are "regular". This is noteworthy for pathological constraint sets, such as integer constraints[7]. For details on $C_{tr}^k(\mathcal{X}, K)$, see (Brudnyi & Brudnyi, 2012a;b) and the extension theorems of Whitney (1934); Fefferman (2005).

**Neural Networks**    It has recently been observed that deep feedforward networks with multiple activation functions, or more generally parametric families of activation functions, achieved significantly more efficient approximation rates than classical feedforward networks with a single activation function (Yarotsky & Zhevnerchuk, 2020; Yarotsky, 2021; Shen et al., 2021a;b). Practically deployed examples of parametric activation functions are the PReLU activation function of He et al. (2015), the Sigmoid-weighted Linear Unit (SiLU) of Elfwing et al. (2018), and the Swish activation function of Ramachandran et al. (2018). We also observe a similar phenomenon, and therefore our quantitative results consider deep feedforward networks whose activation functions belongs to a 1-parameter family $\sigma_\star \triangleq \{\sigma_t\}_{t \in [0,1]} \subseteq C(\mathbb{R})$. The set of all such networks is denoted by $\mathcal{NN}_{n,N}^{\sigma_\star}$ and it includes all $\hat{f} : \mathbb{R}^n \to \mathbb{R}^N$ with iterative representation:

$$\hat{f}(x) \triangleq A^{(J)} x^{(J)}, \qquad x_{i_j}^{(j+1)} \triangleq \sigma_{t_{i_j}}((A^{(j)} x)_{i_j} + b_{i_j}^{(j)}), \qquad x^{(0)} \triangleq x, \qquad (5)$$

where $x \in \mathbb{R}^n$, $j = 1, \ldots, J-1$, each $A^{(j)}$ is a $d_j \times d_{j+1}$-matrix, each $b^{(j)} \in \mathbb{R}^{d_{j+1}}$, $d_{J+1} = N$, $d_1 = 0$, $t_{1,1}, \ldots, t_{J,N_J} \in [0,1]$, for each $j$. The integer $J$ is $\hat{f}$'s *depth* and $\max_{j=1,\ldots,J+1} d_j$ is $\hat{f}$'s *width*.

**Example 1.6** (Networks with Untrainable Nonlinearity). *Denote* $\sigma \triangleq \sigma_0$. *The subset of classical feedforward networks consisting of all* $\hat{f} \in \mathcal{NN}_{n,N}^{\sigma_\star}$ *with each* $\sigma_{t_{i_j}} = \sigma$ *in* (5) *is denoted* $\mathcal{NN}_{n,N}^{\sigma}$.

It is approximation theoretically advantageous to generalize the proposed definition of probabilistic attention in the introduction (3) by replacing $Y$ with a 3-dimensional array (elementary 3-tensor).

**Definition 1.7** (Probabilistic Attention). *Let* $N, Q, m \in \mathbb{N}_+$, *and* $Y$ *be an* $N \times Q \times m$-*array with* $Y_{n,q} \in K$ *for* $n = 1, \ldots, N$, $q = 1, \ldots, Q$. *Probabilistic attention is the function:*

$$\mathbb{R}^n \ni w \mapsto \text{P-attention}(w, Y) \triangleq \frac{1}{Q} \sum_{n=1}^{N} \sum_{q=1}^{Q} \text{Softmax}_N(w)_n \delta_{Y_{n,q}} \in \mathcal{P}_1(K).$$

If $Y$ is an $N \times m$-matrix, as in (3), then we identify $Y$ as the $N \times m \times 1$-array in the obvious manner.

**Set-Valued Analysis:**    A family of non-empty subsets $\{C_x\}_{x \in \mathbb{R}^n}$ of $K$ is said to be a *weakly measurable correspondence*, denoted $C : \mathbb{R}^n \rightrightarrows \mathbb{R}^m$, if for every open subset[8] $U \subseteq K$, $\{x \in \mathbb{R}^n : C_x \cap U \neq \emptyset\}$ is a non-empty Borel subset of $\mathbb{R}^n$ (Aliprantis & Border, 2006, pages 557, 592).

## 2 MAIN RESULTS

We now present our main results in detail. All proofs are relegated to the paper's appendix.

### 2.1 QUALITATIVE APPROXIMATION: DEEP MAXIMUM THEOREM

Our main qualitative result is the following deep neural version of Berge (1963)'s Maximum Theorem where, the measurable selector is approximately implemented by a probabilistic transformer

---

[7]For example, there is no non-constant continuous function $f : [0,1] \to \mathbb{Z}$. However, for any $\lambda \in (0, \frac{1}{2})$ and any $y_1, y_2 \in \mathbb{Z}$, $f = y_1 I_{[0,\lambda]} + y_2 I_{[\lambda + \frac{1}{2}, 1]}$ belongs to $C_{tr}^k([0, \lambda] \cup [\lambda + \frac{1}{2}, 1], \mathbb{Z})$ for each $k \in \mathbb{N}_+$.

[8]Since $K$ is equipped with its subspace topology, then an open subset $U$ of $K$ is any subset of $\mathbb{R}^m$ of the form $U = U_1 \cap K$ where $U_1$ is an open subset of $\mathbb{R}^m$ (see (Munkres, 2000, Lemma 16.1) for further details).

network. We first present the general qualitative result which gives a concrete description of a measurable selector of (1), with high-probability, which has the key property that all its predictions satisfy the required constraints defined by $K$.

**Assumption 2.1** (Kidger & Lyons (2020)). *$\sigma : \mathbb{R} \to \mathbb{R}$ is continuous, $\sigma$ is differentiable at some $x_0 \in \mathbb{R}$, and its derivative satisfies $\sigma'(x_0) \neq 0$.*

**Theorem 2.2** (Deep Maximum Theorem). *Let $\sigma$ satisfy Assumption 2.1. Let $K \subseteq \mathbb{R}^n$ be a non-empty compact set, $C : \mathbb{R}^n \rightrightarrows \mathbb{R}^m$ be a weakly-measurable correspondence with closed values such that $C_x \cap K \neq \emptyset$ for each $x \in \mathbb{R}^n$, $L \in C(\mathbb{R}^m)$, and $\mathbb{P}$ be a Borel probability measure on $\mathbb{R}^n$. For each $0 < \epsilon \leq 1$, there is an $N \in \mathbb{N}_+$, an $\hat{f} \in \mathcal{NN}_{n,N}^\sigma$ of width at most $2 + n + N$, and an $N \times m$-matrix $Y$ such that:*

$$\hat{F} : \mathbb{R}^n \ni x \mapsto \text{P-attention}\left(\hat{f}(x), Y\right) \in \mathcal{P}_1(\mathbb{R}^m), \tag{6}$$

*satisfies the following:*

  *(i)* ***Exact Constrain Satisfaction:*** *$\cup_{x \in \mathbb{R}^n} \text{supp}(\hat{F}(x)) \subseteq K$,*
  *(ii)* ***Probably Approximately Optimality:*** *There is a compact $\mathcal{X}_\epsilon \subseteq \mathbb{R}^n$ satisfying:*
      *(a)* $\max\limits_{x \in \mathcal{X}_\epsilon} \mathscr{W}_1(\hat{F}(x), \operatorname*{argmin}\limits_{y \in C_x \cap K} L(x, y)) \leq \epsilon$,
      *(b)* $1 - \mathbb{P}(\mathcal{X}_\epsilon) \leq \epsilon$.

Theorem 2.2 implies that for any random field $(Y^x)_{x \in \mathbb{R}^n}$ on $\mathbb{R}^m$ (i.e. a family of $\mathbb{R}^m$-valued random vectors indexed by $\mathbb{R}^n$) with $Y^x \sim \hat{F}(x)$: 1. samples drawn from $Y^x$ are in $K$ (by (i)) and 2. samples drawn from each $Y^x$ are near to the optimality set $\operatorname{argmin}_{y \in C_x \cap K} L(x, y)$ (by (ii)).

**Corollary 2.3** ($\hat{F}$'s Mean Prediction). *Assume the setting of Theorem 2.2. Let $\{Y^x\}_{x \in \mathbb{R}^n}$ be a $K$-valued random field with $Y^x \sim \hat{F}(x)$ for each $x \in \mathbb{R}^n$ then, $1 - \mathbb{P}(\mathcal{X}_\epsilon) \leq \epsilon$ and*

$$\max_{x \in \mathcal{X}_\epsilon} \mathbb{E}[\|Y^x - \operatorname*{argmin}_{y^\star \in C_x \cap K} L(x, y^\star)\|] \leq \epsilon.$$

Appendix 8 contains additional consequences of the Deep Maximum Theorem, such as the special case of classical transformers when $K$ is convex. Next, we complement our qualitative results by their quantitative analogues, within the context of *universal approximation* under *constraints*.

## 2.2 Quantitative Approximation: Constrained Universal Approximation

In order to derive a *quantitative* constrained universal approximation theorem, we require the loss function to be tied to the Euclidean norm in the following manner.

**Assumption 2.4** (Norm-Controllable Loss). *There is a continuous $f : \mathbb{R}^n \to \mathbb{R}^m$ with $f(\mathbb{R}^n) \subseteq K$ and a continuous $l : [0, \infty) \to [0, \infty)$ with $l(0) = 0$, satisfying: $L(x, y) \leq l(\|f(x) - y\|)$.*

Just as with transformer networks, our "constrained universal approximation theorem" approximates a suitably regular function $f : \mathbb{R}^n \to K \subseteq \mathbb{R}^m$ while exactly respecting the constraints $K$ by implementing an *encoder-decoder* network architecture. Thus, our model is a composition of an *encoder network* $\hat{\mathcal{E}} : \mathbb{R}^n \to \mathbb{R}^d$ whose role is to approximate $f$ in a classical "unconstrained fashion" and a *decoder network* (with probabilistic attention layers at its output) $\hat{\mathcal{D}} : \mathbb{R}^d \to \mathcal{P}_1(K)$ whose role is to enforce the constraints $K$ while preserving the approximation performed by $\hat{\mathcal{E}}$, where $d \lll m$.

To take advantage of the encoder-decoder framework present in most transformer networks, we formalize what is often called a *"latent low-dimensional manifold"* hypothesis. Briefly, this means that, the hard constraints in set $K$ are *contained* in a "low dimensional" subspace.

**Assumption 2.5** (Low-Dimensional Manifold). *There is an $0 < s$ and a smooth bijection $\Phi$ from $\mathbb{R}^n$ to itself with smooth inverse[9], such that $\Phi(K) \subseteq \mathbb{R}^d$; where $2 \leq d$ and $d \in \Theta(m^{\frac{1}{s}})$.*

Assumption 2.5 does not postulate that $K$ is itself a single-chart low-dimensional manifold, or even a manifold. Rather, $K$ need only be contained in a low-dimensional manifold. For the fast rates we use activation functions generalizing the swish function (Ramachandran et al., 2018) as follows.

**Assumption 2.6** (Swish-Like Activation Function). *The map $\sigma : [0, 1] \times \mathbb{R} \ni (\alpha, t) \mapsto \sigma_\alpha(t) \in \mathbb{R}$ is continuous; $\sigma_0$ is non-affine and piecewise-linear, and $\sigma_1$ is smooth[10] and non-polynomial.*

---

[9] Here smooth means that $\Phi$ is continuously differentiable any number of times. NB, smooth bijections with smooth inverses are called *diffeomorphisms* in the differential geometry and differential topology literature.

[10] Following Jost (2017), a function $\sigma : \mathbb{R} \to \mathbb{R}$ is called smooth (or class $C^\infty$) if $\partial^k \sigma$ exists for each $k \in \mathbb{N}_+$.

**Theorem 2.7** (Constrained Universal Approximation). *Let $k \in \mathbb{N}_+$ and $\mathcal{X} \subseteq [0,1]^n$ be non-empty. Suppose that $\sigma$ satisfies 2.6, $L$ satisfies Assumption 2.4, $K \subseteq \mathbb{R}^n$ is non-empty, compact and satisfies Assumption 2.5. For any $f \in C_{tr}^k(\mathcal{X}, K)$, every constraining quality $\epsilon_K > 0$, and every approximation error $\epsilon_f > 0$, there exist $N, Q \in \mathbb{N}_+$, an encoder $\hat{\mathcal{E}} \in \mathcal{NN}_{n,d}^\sigma$, and a decoder:*

$$\hat{\mathcal{D}} : \mathbb{R}^d \ni x \mapsto \sum_{k=1}^N \text{P-attention}\left(\hat{D}(x), Y\right) \in \mathcal{P}_1(K) \qquad (7)$$

*where $\hat{D} \in \mathcal{NN}_{d,N}^\sigma$ and $Y$ is an $N \times Q \times m$-array with $Y_{1,1}, \dots, Y_{N,Q} \in K$ such that:*

(i) ***Exact Constrain Satisfaction:*** *For each $x \in \mathbb{R}^n$: $\text{supp}(\hat{\mathcal{D}} \circ \hat{\mathcal{E}}(x)) \subseteq K$,*
(ii) ***Universal Approximation:*** *The estimate holds [11]:*

$$\sup_{x \in [0,1]^n} \mathscr{W}_1(\hat{\mathcal{D}} \circ \hat{\mathcal{E}}(x), \operatorname*{argmin}_{y \in K} L(x,y)) \le \epsilon_K + k \operatorname{Lip}(\Phi^{-1}) d\epsilon_f;$$

*where, $0 < k$ is an absolute constant independent of $n$, $m$, $d$, $f$, and of $\epsilon$ and $\operatorname{Lip}(\Phi^{-1})$ denotes the Lipschitz constant of $\Phi^{-1}$ on the compact set $\{z \in \mathbb{R}^d : \|z - \Phi(K)\| \le \epsilon_f\}$.*

*Furthermore, the "complexities" of $\hat{\mathcal{D}}$ and $\hat{\mathcal{E}}$ are recorded in Table[12] 1 for $\frac{\epsilon}{2} = \epsilon_k = \epsilon_f$.*

In practice, we can only *sample* from each measure $\hat{\mathcal{D}} \circ \hat{\mathcal{E}}(x)$. In this case, we may ask how the typical sample drawn from a random-vector $Y^x$ distributed according to our learned measure $\hat{\mathcal{D}} \circ \hat{\mathcal{E}}(x)$ performs when minimizing $L(x, y)$. The next result relates the estimates in Theorem 2.7 (ii) to the typical (in $Y^x$) worst-case (in $x$) gap between a sample from $Y^x$ and $f(x)$, as quantified by $L(x, \cdot)$.

**Corollary 2.8** (Average Worst-Case Loss). *Assume the setting of Theorem 2.7 and suppose that the "modulus" $l$ in Assumption 2.4 is strictly increasing and concave. Let $\hat{\mathcal{D}}$ and $\hat{\mathcal{E}}$ be as in Theorem 2.7 and let $\{Y^x\}_{x \in \mathcal{X}}$ be an $\mathbb{R}^m$-valued random field with $Y^x \sim \hat{\mathcal{D}} \circ \hat{\mathcal{E}}(x)$, for each $x \in \mathbb{R}^n$. Then:*

$$\max_{x \in \mathcal{X}} \mathbb{E}_{Y^x \sim \hat{\mathcal{D}} \circ \hat{\mathcal{E}}(x)}[L(x, Y^x)] \le l\left(\epsilon_K + k \operatorname{Lip}(\Phi^{-1}) d\epsilon_f\right).$$

Corollary 2.8 quantifies the expected performance of a sample from our probabilistic transformer model, as expressed by $L$, whereas Theorem 2.7 (ii) quantifies the difference from the transformer's prediction to the optimal prediction value. Next, we consider implications of our main results.

## 2.3 APPLICATIONS

We apply our theory to obtain a universal approximation theorem for classical transformer networks with exact convex constraint satisfaction and to derive a version of the non-Euclidean universal approximation theorems of Kratsios & Bilokopytov (2020); Kratsios & Papon (2021) for Riemannian-manifold valued functions which does not need explicit charts. As with most quantitative (uniform) universal approximation theorems (Gühring et al., 2020; Kidger & Lyons, 2020; Shen et al., 2021a), we henceforth consider $L(x, y) = \|f(x) - y\|$. We also fix $f \in C_{tr}^k([0,1]^n, K)$.

### 2.3.1 TRANSFORMERS ARE CONVEX-CONSTRAINED UNIVERSAL APPROXIMATORS

We return to the familiar transformer networks of Vaswani et al. (2017). The next result shows that transformer networks can balance universal approximation and exact convex constraint satisfaction. This is because when $K$ is convex, then the mean of the random field $\{Y^x\}_{x \in \mathbb{R}^n}$ of Corollary 2.3 must belong to $K$. Consequently, the identity (4) implies that $\text{Attention}(\hat{\mathcal{D}} \circ \hat{\mathcal{E}}(\cdot), Y) \approx f$.

**Corollary 2.9** (Constrained Universal Approximation: Convex Constraints). *Consider the setting and notation of Corollary 2.8. Suppose that $K$ is convex and let $L(x, y) = \|f(x) - y\|$. Then:*

$$\mathbb{R}^n \ni x \mapsto \mathbb{E}[Y^x] = \text{Attention}(\hat{\mathcal{D}} \circ \hat{\mathcal{E}}(x), Y) \in K; \qquad (8)$$

(i) ***Exact Constraint Satisfaction:*** *$\mathbb{E}_{Y^x \sim \hat{\mathcal{D}} \circ \hat{\mathcal{E}}(x)}[Y^x] \in K$, for each $x \in \mathbb{R}^n$,*
(ii) ***Universal Approximation:*** *$\sup_{[0,1]^n} \|f(x) - \mathbb{E}_{Y^x \sim \hat{\mathcal{D}} \circ \hat{\mathcal{E}}(x)}[Y^x]\| < \epsilon_K + kd\epsilon_f$.*

*The "complexities" of the networks $\hat{\mathcal{D}}$ and $\hat{\mathcal{E}}$ are recorded in Table [13] 1 for $\frac{\epsilon}{2} = \epsilon_k = \epsilon_f$.*

---

[11]In fact, we actually prove that the slightly stronger statement: $\sup_{x \in [0,1]^n} \mathscr{W}_1\left(\hat{\mathcal{D}} \circ \hat{\mathcal{E}}(x), \delta_{f(x)}\right) \le \epsilon_K + k \operatorname{Lip}(\Phi^{-1}) d\epsilon_f$. Both formulations align when $l$ has a unique minimum at 0, as is the case when $L(x, y) = \|f(x) - y\|_\star$ and $\|\cdot\|_\star$ is any norm on $\mathbb{R}^m$.

[12]Explicit constants are recorded in Table 2 within the paper's appendix; there, $\epsilon_K$ and $\epsilon_f$ may differ.

[13]Explicit constants are recorded in Table 2 within the paper's appendix; there, $\epsilon_K$ and $\epsilon_f$ may differ.

### 2.3.2 CHART-FREE RIEMANNIAN MANIFOLD-VALUED UNIVERSAL APPROXIMATION

We explore how additional non-convex structure of the constraint set $K$ can be encoded by the *probabilistic transformer networks* of Theorems 2.2 and 2.7 and be used to build new types of (deterministic) transformer networks. These results highlight that the standard transformer networks of (8) are specialized for convex constraints and that by instead using an intrinsic variant of expectation, we build can new types of "geometric transformer networks" customized to $K$'s geometry. This section makes use of Riemannian geometry; for an overview see Jost (2017).

Let $(M, g)$ be a connected $d$-dimensional Riemannian submanifold of $\mathbb{R}^m$ with distance function by $d_g$. We only require the following mild assumption introduced in Afsari (2011). We recall that the *injectivity radius* at $y_0$, denoted by $\inf_g(y_0)$, (see (Jost, 2017, Definition 1.4.6)) is the minimum length of a *geodesic* (or minimal length curve) in $M$ with starting point $y_0$. We also recall that the *sectional curvature* (see (Jost, 2017, Definition 4.3.2) for a formal statement) quantifies the curvature of $(M, g)$ as compared the geometry of its flat counterpart $\mathbb{R}^d$. We focus on a broad class of non-convex constrains, namely *geodesically convex constraints*, which generalize convex constraint and have received recent attention in the optimization literature (Zhang & Sra, 2016; Liu et al., 2017).

**Assumption 2.10** (Geodesically Convex Constraints). *The Riemannian manifold $(M, g)$ is connected, it is complete as a metric space, and all its sectional curvatures of $(M, g)$ are all bounded above by a constant $C \geq 0$. The non-empty constrain set $K$ satisfies:*

1. *$K$ is contained in the geodesic ball $B(y_0, \rho) \triangleq \{y \in M : d_g(y_0, y) < \rho\}$ for some point $y_0 \in M$ and some radius $\rho$ satisfying[14]: $0 < \rho < 2^{-1} \min\{\text{inj}_g(y_0), \frac{\pi}{\sqrt{C}}\}$,*
2. *For each $y_0, y_1 \in K$ there exists a unique geodesic $\gamma : [0, 1] \to K$ joining $y_0$ to $y_1$.*

Our latent probabilistic representation grants us the flexibility of replacing the usual "extrinsic mean" used in (8) to extract deterministic predictions from our probabilistic transformer networks via an additional *Fréchet mean* layer at their readout. This intrinsic notion of a mean, was introduced independently in Fréchet (1948) and in Karcher (1977), and is defined on any $\mathbb{P} \in \mathcal{P}_1(K)$ by:

$$\bar{\mathbb{P}} \triangleq \underset{k \in K}{\text{argmin}} \int d_g^2(k, u) \, \mathbb{P}(du). \tag{9}$$

With this "geometric readout layer" added to our model, we obtain the following variants of our main results in this non-convex, but geometrically regular, setting.

**Corollary 2.11** (Constrained Universal Approximation: Riemannian Case). *Consider the setting and notation of Corollary 2.8. Let $L(x, y) = \|f(x) - y\|$. If Assumption 2.10 holds then:*

$$\mathbb{R}^n \ni x \mapsto \overline{\hat{\mathcal{D}} \circ \hat{\mathcal{E}}(x)} \in K, \tag{10}$$

*is a well-defined Lipschitz-continuous function, and the following hold:*

(i) *Exact Constraint Satisfaction: $\overline{\hat{\mathcal{D}} \circ \hat{\mathcal{E}}(x)} \in K$, for each $x \in \mathcal{X}$,*

(ii) *Universal Approximation: $\sup_{\mathcal{X}} d_g(f(x), \overline{\hat{\mathcal{D}} \circ \hat{\mathcal{E}}(x)}) < \epsilon_K + kd\epsilon_f$.*

*The "complexities" of $\hat{\mathcal{D}}$ and $\hat{\mathcal{E}}$ are recorded in Table[15] 1 for $\frac{\epsilon}{2} = \epsilon_k = \epsilon_f$.*

## 3 DISCUSSION

In this paper, we derived the first *constrained universal approximation* theorems using probabilistic reformation of Vaswani et al. (2017)'s transformer networks. The results assumed both a quantitative form (Theorem 2.7) and a qualitative form in the more general case of an arbitrary loss functions $L$ and additional compatible soft constraints in (Theorem 2.2). Our results provide (generic) direction to end-users designing deep learning models processing non-vectorial structures and constraints.

As this is the first approximation theoretic result in this direction, there are naturally as many questions raised as have been answered. In particular, it is natural to ask: "*Are the probabilistic transformer networks trainable in practice; especially when $K$ is non-convex?*". In Appendix 5, we show that the answer is indeed: "*Yes!*", by proposing a training algorithm in that direction and showing that we outperform an MLP model and a classical transformer network in terms of a joint MSE and distance to the constraint set. The evaluation is performed on a large number of randomly generated experiments, whose objective is to reduce the MSE to a randomly generated function mapping a high-dimensional Euclidean space to there sphere $\mathbb{R}^3$ with outputs constrained to the sphere.

---

[14]Following Afsari (2011), we make the convention that if $C \leq 0$ then, $\frac{1}{\sqrt{C}}$ is interpreted as $\infty$.

[15]Explicit constants are recorded in Table 2 within the paper's appendix; there, $\epsilon_K$ and $\epsilon_f$ may differ.

## ACKNOWLEDGMENTS

Anastasis Kratsios and Ivan Dokmanić were supported by the European Research Council (ERC) Starting Grant 852821—SWING. The authors thank Wahid Khosrawi-Sardroudi, Phillip Casgrain, and Hanna Sophia Wutte from ETH Zürich, Valentin Debarnot from the University of Basel for their helpful feedback, and Sven Seuken from the University of Zürich for his helpful feedback in the rebuttal phase.

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
