# OpenReview forum: "Universal Approximation Under Constraints is Possible with Transformers"
_ICLR.cc/2022/Conference — ICLR 2022 Spotlight_

### Official Review · Reviewer_tJ9y · 2021-11-02

**Correctness:** 4
**Technical Novelty And Significance:** 4
**Empirical Novelty And Significance:** Not applicable
**Recommendation:** 10
**Confidence:** 4

**Main Review:**

### Strengths

**Examples.**  Throughout the paper, the authors give numerous examples to illustrate why and how one might have constraints when training neural networks.  These were quite helpful in giving intuition for the (rather technical) contents of the paper.  This is a model I wish more technical papers would follow, as it allows even non-technical readers to get an idea of what the theory is all about.  Further, the authors do a commendable job of relating this work to the statistical results which have recently been proposed.

**Problem statement & setting.**  The problem statement is really clearly articulated.  The idea of providing a universal approximation property under constraints seems quite natural, and the authors motivate it extremely well.  Indeed, it is true that there are many works that argue for constraints in deep learning without considering the question of whether satisfying such constraints is even possible.

**Technical quality.**  I was not able to read through all of the proofs, especially given that the appendix gets pretty involved.  However, from what I saw, the proofs are of high quality.  Indeed, the generalization to the case of nonconvex K seems quite involved.  The proof techniques used here should be of wider interest.

**Informal theorem presentation.**  The choice to first state the main results informally greatly helped the readability of the paper.  The main results are also illustrated with useful diagrams.

**Novelty.**  This is the first result of this kind that I am aware of.  For this reason, I believe that this paper has a significant claim to novelty.

**Experiments.**  I like the experiments at the end of the appendix as well.  This helps to cover another natural question regarding how one can use these results in practice.

### Weaknesses

**On the attention module.**  I think the paper could benefit from further explaining why this particular form of attention is natural.  It would also perhaps help to illustrate this with an example.  As the credibility of the paper relies on the reader accepting that this attention model is natural and general enough to form the basis of the approach to universal approximation, I think that it's worth spending some more time here.  In particular, I didn't get a good sense of the role of the matrix $Y$ in build up to Informal Theorem 1.

**Figures.**  Don't get me wrong, I love that there are figures here.  However, I think that they could be explained better.  For instance, it wasn't clear to me what the green region was in Figure 1.  Is this $K$?  Also, the blue dot seems too small, and thus quite hard to see.

**Summary Of The Paper:**

This paper presents a family of constrained universal approximation results for probabilistic transformers.  The authors provide significant theoretical contributions for both convex and non-convex constraint sets.  In my opinion, this represents a significant advance in our understanding of universality in ML.

**Summary Of The Review:**

My review for this paper is rather short because overall I think that this paper is excellent.  It's clear that the problem is well-motivated and there are few results in the area.   To this end, the authors provide highly technical results to specifically address this novel and important question.  Moreover, the authors make a great effort to explain their results to a non-technical audience.  This is a clear accept in my opinion, and should be highlighted at the conference.

---

> ### Public Comment · ~Stella_Biderman1 · 2021-11-10
> **$Y = V$**
>
> The standard formulation of attention is the dot product attention: $\mathop{softmax}\left(QK^T\right)V$, where $Q$ is known as the query, $K$ is known as the key, and $V$ is known as the value. This is what is used in (among other places), "BERT: Pre-training of Deep Bidirectional Transformers for
> Language Understanding," "Language Models are Few-Shot Learners," "Exploring the Limits of Transfer Learning with a Unified Text-to-Text Transformer," and "Multitask Prompted Training Enables Zero-Shot Task Generalization." In the paper under review, the authors set $QK^T=w$ and $V=Y$.
>
> Can you explain what other formulation of attention you view as being commonly used? While it's not the case that every single paper uses dot product attention, even among papers that don't they are typically using another formulation that still has a "value" matrix or are approximating the above equation.

---

> > ### Comment · Reviewer_tJ9y · 2021-11-10
> > **Thanks for clarifications regarding attention + elaboration on my review**
> >
> > Thanks for your comment and the clarifications about attention; that's super helpful!  Perhaps I should have been more specific.  My point is not that there may be an alternative definition of attention that should be used here.  Rather, my comments stem from the my perception that while (as you point out) attention seems to be a commonly use mechanism in domains like NLP, it is less common for other tasks such as image classification.  Now all cards on the table: I'm not an expert in NLP.  So perhaps my (minor) criticism of the lack of motivation for attention is without merit for NLP.  However, as I read the paper, I noticed that many of the references that were used to motivate the need for constraints in learning were ones in which (I believe) attention is less standard; a sampling of these references from the paper:
> >
> > Lukas Gonon, Lyudmila Grigoryeva, and Juan-Pablo Ortega. Approximation bounds for random
> > neural networks and reservoir systems. arXiv preprint arXiv:2002.05933, 2020b.
> >
> > Alexander Robey, George J Pappas, and Hamed Hassani. Model-based domain generalization.
> > arXiv:2102.11436, 2021. URL https://arxiv.org/abs/2102.11436.
> >
> > Michael M Bronstein, Joan Bruna, Yann LeCun, Arthur Szlam, and Pierre Vandergheynst. Geometric deep learning: going beyond Euclidean data. IEEE Signal Processing Magazine, 34(4):18–42,
> > 2017.
> >
> > Chi Jin, Praneeth Netrapalli, and Michael Jordan. What is local optimality in nonconvex-nonconcave
> > minimax optimization? In Proceedings of the International Conference on Machine Learning
> > (ICML), 2020.
> >
> > None of these works mention/use attention.  So I think that given that the authors motivate their results with these kinds of applications in mind, there should be some more space devoted to motivating the attention mechanism for these tasks.  If you disagree, I'm more than happy to discuss further.  (But at the end of the day, as you can tell from my score, I'm more than happy to see this paper be accepted!)

---

> > > ### Public Comment · ~Stella_Biderman1 · 2021-11-11
> > > **Fair enough, given the context**
> > >
> > > That’s a pretty good point. As an expert in large transformer models, I skipped over that bit and generally take the interestingness of attention for granted.
> > >
> > > Given that this is a theory paper for a general audience conference, I agree it’s not something the paper should necessarily take for granted. It would probably behoove this paper to cite some NLP work (e.g., BERT, GPT-2, GPT-3, T5, GPT-J) as an example application where this model of computation is very interesting. My work with vision models so far has largely been limited to multimodal image-text model, and if in application areas that don’t source to NLP typically use a different form of attention then that’s absolutely something worth acknowledging and discussing.

---

> > > > ### Author Response · Authors · 2021-11-12
> > > > **A new spin on attention for solving geometric machine learning problems**
> > > >
> > > > The interpretation of $Y$ in our setting is an array of "particles" on the constraint set $K$.  There are certainly many problems successfully addressed by our proposed versions of attention, though those problems may differ from those of focus in MLP applications.  Especially concerning our question: *"Can one approximate a $K$-valued function while exactly encoding the constraint defined by $K$?"*, our attention interpretation is appealing since it is simple, can easily be written down in practice, and allows us to leverage probabilistic tools.

---

> ### Author Response · Authors · 2021-11-12
> **Thank you for the strong endorsement! (And further motivations for the used model)**
>
> Thank you for taking the time to review our paper. We are very pleased that you enjoyed it!
>
> *On the attention module. I think the paper could benefit from further explaining why this particular form of attention is natural. It would also perhaps help to illustrate this with an example. As the credibility of the paper relies on the reader accepting that this attention model is natural and general enough to form the basis of the approach to universal approximation, I think that it's worth spending some more time here. In particular, I didn't get a good sense of the role of the matrix in build up to Informal Theorem 1.*
>
> **Response:**
> This is a very good point. We hope that the additions in the updated manuscript and the appendices address it. We expanded the discussion following Figure $1$ and added a discussion in the new Appendix B.3 section (integrated in page 14 following Figures $6$ and $7$) which explains why this form of attention is natural for our problem.  We also included additional numerical illustrations in the same appendix, which we hope help convince the reader of the suitability of this form of attention to the problem of approximation under constraints, for constraints ranging between convex and highly non-convex. In Figure $1$, when $K$ is a convex polytope the ``particles'' $Y_1,\dots,Y_N$ may be placed on $K$'s vertices.
>
> *Figures. Don't get me wrong, I love that there are figures here. However, I think that they could be explained better. For instance, it wasn't clear to me what the green region was in Figure 1. Is this?*
> *Also, the blue dot seems too small, and thus quite hard to see. *
>
> **Response:**
> Based on your suggestions, as well as those of other reviewers, we expanded the figure descriptions in the text.  The green part of Figure 1 represents $K$, the purple dots represent the ``particles'' $Y_1,\dots,Y_N$, the size of the violet bubble around each particle illustrates the probability that it is predicted by the probabilistic transformer, and the blue dot is the weighted average of the particles.  Thus, the blue dot (which we enlarged) corresponds to the output of the deterministic transformer.  Moreover, we took the opportunity to redraw Figures 3 and 4; they should be clearer now.
>
> Thank you again for your encouraging assessment and helpful feedback.

---

### Official Review · Reviewer_w3Aa · 2021-11-03

**Correctness:** 3
**Technical Novelty And Significance:** 3
**Empirical Novelty And Significance:** 1
**Recommendation:** 6
**Confidence:** 4

**Main Review:**

Strength: + Provides rigorous theory to support the utility of transformers for deep neural network based predictive purposes under constrained settings.
+ Soft constraint set that is defined using available data points is an interesting aspect that the paper considers. In fact, the authors were able to extend their theory to show that transformers can be used to model any data dependent constraint sets as well.
+ Results of this paper may be applicable to practical use cases.
Weakness: - The paper is hard to follow at many places since only informal theorems are provided throughout the paper.
- The main theorem 2.2 is suboptimal -- optimal network may require width greater than the number of input dimensions n. For image analysis, this corresponds to the number of pixels.
-Confusing notations.
- No experiments to verify any of the quantitative results.


**Summary Of The Paper:**

The paper intends to provide a stronger type of universal approximation result of transformer models. However, more details on experiments is required with a major revision of presentation and clarity.

**Summary Of The Review:**

Justification: The constraint examples are too unnatural. For example, in geometric deep learning, authors argue that when K is a disjoint set, then the closest point computation becomes an unmanageable dilemma, claiming that the linear relaxations are meaningless for integer program (IP) while LPs are routinely used to solve IPs in practice using a branch and bound scheme. While some pieces of notations are kept constant throughout the paper, there are many confusing notations at various places that can easily be avoided - "Attention" in eq (4) is different from "attention". Adding to this issue, another central issue with the presentation is that all the technical statements in the main paper are Informal theorems with no rigorous explanations provided. This makes it very hard to read the paper because while the statements themselves may be locally correct or correct in isolation, it sometimes does not even make sense when put together as a whole, and makes it possible to verify the technical correctness of the paper. Somewhat related to the previous point, some technically deep concepts (such as Wasserstein-1 distance) are introduced in passing and handled in a slightly cavalier manner. It could have been nice if the authors provided some explanations to the figures -- at the moment, the reader has to guess what the authors intend to say.

Given that the paper shows quantitative results in the infinite sample setting, it is not clear whether the results hold in the finite sample setting where transformers are being used in practice. It would have been nice if the authors can formulate and show some toy experiments to even check whether such universality may be possible for dataset in the hands of a practitioner.

After response: Thanks for the detailed response. I agree with the modifications, revisions, additional sections, and references proposed in the response. These changes were really helpful in parsing the paper for me since I have adequate mathematical background, but not so much experience in theoretical computer science. With more clearer notations, and figure descriptions, I can now see that the paper indeed focuses on universal approximation rather than generalization, and now I feel that my criticism about finite sample settings has been answered by the toy experiments also.  I choose the option of "minor issues" in correctness only because some statements still read handwavy in Section 1, for example, there are many places where the authors use the word "any", as in, any loss function, any constraint set, any goood activation etc.. While it may be correct in some sense, I find it odd to use in an otherwise rigorous paper. As an aside, I would like to point out this certainly does not hold if one wants to train from scratch since we know that loss functions and activation functions have a significant effect on first order algorithms. Thanks for providing additional references for us to check, and while going through them I found that some of the theoretical papers have very relevant practical use cases and I believe that they may add significant value to the overall scope and presentation given here, for example, Meta-learning with implicit gradients. I have raised my score.

---

> ### Author Response · Authors · 2021-11-12
> **Additional computer experiments (but to be interpreted with care!)**
>
> **[Multipart response (part 3/3)]**
>
> *No experiments to verify any of the quantitative results...It would have been nice if the authors can formulate and show some toy experiments to even check whether such universality may be possible for dataset in the hands of a practitioner.*
>
> **Response:**
> We agree that stylized numerical experiments may help validate our quantitative theoretical claims. In the new Section B.3 we set out to probe the main new quantitative statement in Theorem 2.2. Namely, we confirm that ``The model complexities in Table 1 are independent of $K$'s geometry''. We do so by approximating various functions taking values in a constraint set whose geometry is progressively made more complicated, beginning with a polytope, moving to a nonpolytopal convex set, then to a manifold, and eventually to a "degenerate" non-convex set which is not a manifold.  The depth and width of the approximating architecture are fixed across all experiments in order. We find that the probabilistic transformer implementing the approximation indeed performs comparably across all geometries relative to the MLP.
>
> Let us add that, a priori, our results guarantee existence of a choice of weights such that the proposed network approximates any admissible target function with a desired accuracy, but they do not tell us how to find those weights. How to do this with guarantees is largely an open question at the interface of statistics and (non-convex) optimization theory. Such empirical verifications (including ours) thus have to be interpreted with care and are therefore uncommon in papers on universal approximation (e.g., Kidger et al. (COLT 2020) “Universal Approximation with Deep Narrow Networks,” S. Park et al. (ICLR 2021) “Minimum Width for Universal Approximation”, R. Gribonval et al. (Constructive Approximation 2021) “Approximation spaces of deep neural networks”, I. Gühring and M. Raslan (2021 Neural Networks) “Approximation rates for neural networks with encodable weights in smoothness spaces”, etc.).

---

> > ### Comment · Reviewer_tJ9y · 2021-11-16
> > **This review amplifies flaws while downplaying the significance.  Could the reviewer elaborate more?**
> >
> > I just wanted to add my two cents here.  I agree with the authors regarding many of the points raised in Reviewer w3Aa's review.  In particular, I agree with the authors that their results are quite general WRT the kinds of constraints that their theory covers.  Indeed, in my view, the results concerning convex constraints would be worthy of publication by themselves.  Thus, the fact that the authors were able to extend these results to certain non-convex sets is quite impressive and noteworthy.
> >
> > Furthermore, I think we should discuss further why Reviewer w3Aa feels that experiments are needed here.  This is a paper about universal approximation.  The goal of such a theory is not to be of immediate practical use; rather, the theory is meant to show that certain classes of NNs can satisfy some property (in this case the NN satisfy a set of constraints $K$).  To be clear, the theory doesn't tell us anything about **how** one might go about finding such a NN. Moreover, I think we should acknowledge that there is a **significant theoretical contribution** here.  In this way, I disagree that the paper should be criticized for not providing experiments.  Indeed, many papers are accepted at ICLR which do not have contain experiments; this is not a weakness of the paper.  So my question to Reviewer w3Aa is: Why do you feel that experiments are needed here?
> >
> > I also agree with the authors regarding the finite sample guarantees mentioned in Reviewer w3Aa's review.  I don't believe that the authors made any claims about the finite sample setting, and therefore I don't understand why Reviewer w3Aa listed this as a weakness.  Perhaps the reviewer could elaborate on this more?
> >
> > Overall, I feel that this review is rather harsh in the sense that it downplays the significance of the results while simultaneously amplifying various flaws, most of which (in my mind) are relatively minor or non-applicable (as discussed above).  Ultimately, our job as reviewers is to be critical, but not to the point where we are rejecting work that is clearly strong and useful because of minutiae.  Furthermore, if there are flaws, it seems to be that as reviewers we should be responsible for identifying them in a specific way in an effort to improve the paper.   I feel that this review has not done that, given that many of the statements are vague and do not give the authors constructive feedback. This being the case, I think it would be appropriate for Reviewer w3Aa to elaborate more on the points raised in their review.  Or, if the response from the authors was satisfactory toward addressing the points raised in the review, I think that the reviewer should consider raising their score.

---

> > > ### Comment · Reviewer_w3Aa · 2021-11-30
> > > **Need some more time to read**
> > >
> > > Dear Reviewer tJ9y,
> > >
> > > Thanks for actively championing! It has been a really busy couple of weeks, especially with the semester coming to close at where I'm employed. So, I need one more week to reread the revision and provide an overall rating to the paper that also is well calibrated with respect to the other theoretical papers (evidenced with sufficient synthetic/toy experiments) that I am reviewing this same year at ICLR. I went through your comments, other reviews, and authors' feedback in brief. I feel like their revision addresses some of the concerns in my initial review, and to that end, I will revise my score also.

---

> > > > ### Author Response · Authors · 2021-12-08
> > > > **Curious to hear feedback**
> > > >
> > > > Dear reviewer w3Aa,
> > > >
> > > > Thank you for your willingness to read our discussions and the updated manuscript even after the discussion period has ended—we were delighted to read your response to reviewer tJ9y. We would like to know whether the new explanations and updates have changed your assessment of the paper.
> > > >
> > > > Best,
> > > >
> > > > Authors

---

> > > > > ### Comment · Reviewer_w3Aa · 2021-12-09
> > > > > **Thanks for the response**
> > > > >
> > > > > I have updated my review. The updated manuscript and response answers most of the critical points in my initial review.

---

> ### Author Response · Authors · 2021-11-12
> **Our results cover a great variety of constraint sets, from integers, to manifolds, to rather strange sets**
>
> **[Multipart response (part 2/3)]**
>
>
> *[I]n geometric deep learning, authors argue that when K is a disjoint set, then the closest point computation becomes an unmanageable dilemma, claiming that the linear relaxations are meaningless for integer program (IP) while LPs are routinely used to solve IPs in practice using a branch and bound scheme.*
>
> **Response**
> We use disconnected sets as an example of extremes covered by our theory; our main geometric deep learning results concern the problem of approximating functions that take values in a Riemannian manifold (Section 2.3.2, esp. Corollary 2.11). These results show that when the constraint set $K$ is a geodesically convex subset of a (complete) Riemannian manifold, universal approximation of a $K$-valued function is possible with a transformer which takes values in $K$. We emphasize that, unlike the motivational discussion about integers, our results about manifolds are of central interest in geometric deep learning (e.g., the survey papers: M. Bronstein et al. (2017) "Geometric deep learning: going beyond Euclidean data" Section 4 and Section 4.4 of M. Bronstein et al. (2021) " Geometric Deep Learning: Grids, Groups, Graphs, Geodesics, and Gauges"). Furthermore, our results are the only available chart-free guarantees (Corollary 2.11) in this direction.
>
> You are right that LPs, perhaps combined with branch-and-bound, may be used to approximate (or exactly solve) certain mixed-)integer programs. But our paper is (among many other things) about universally approximating solution maps to such constrained problems by neural networks (e.g. Baes et al. Low-rank plus sparse decomposition of covariance matrices using neural network parametrization" (2021), Cai et al. (2021) "Learned Robust PCA: A Scalable Deep Unfolding Approach for High-Dimensional Outlier Detection"). The integer example serves only to illustrate a technical obstruction whose removal requires a new mathematical idea. We give first existing results in this direction. We emphasize that integer constraints are only a very specific instance of the broad class of constraints handled by our results (from familiar constraints like polytopes to bizarre constraints like disjoint unions of sets of different dimension), most of which do not admit any known mathematical programming technique.
>
> *[T]here are many confusing notations at various places that can easily be avoided - "Attention" in eq (4) is different from "attention".*
>
> **Response:**
> With this notation we tried to emphasize the strong link between the two concepts (Equation (4)), but we now realize that it may be quite confusing. We thus replaced ``attention'' by $P-\mathrm{attention}$. We also scanned the manuscript for other possibly confusing notation. If there is anything else you find problematic please let us know.
>
> *Given that the paper shows quantitative results in the infinite sample setting, it is not clear whether the results hold in the finite sample setting where transformers are being used in practice.*
>
> **Response:**
> Our paper indeed derives a universal approximation guarantee and not a statistical guarantee (such as a bound on the generalization gap). In this sense, our main statement adds to the line of work spearheaded by Hornik et al. in "Multilayer feedforward networks are universal approximators" (1989) and Cybenko "Approximation by superpositions of a sigmoidal function" (1989) and parallels recent sophisticated versions for networks on graphs (Keriven and Peyre, "Universal Invariant and Equivariant Graph Neural Networks" (NeuIPS 2019)), networks that obey symmetries specified by certain groups (Yarotsky "Universal approximations of invariant maps by neural networks" (Constructive Approximation, 2021)), etc. Statistical guarantees are important but complementary to our results and they require a rather different set of techniques; see for instance: Alquier et al. "On the properties of variational approximations of {G}ibbs posteriors" (2016) or Ribeiro et al. "Probably approximately correct constrained learning" (2020).

---

> ### Author Response · Authors · 2021-11-12
> **The constraint examples are common in machine learning and signal processing**
>
> **[Multipart response (part 1/3)]**
>
> We thank the reviewer for their valuable feedback.  We respectfully disagree with most of the critical points raised (and hope to articulate why below), but we feel that many of those helped us improve the manuscript.
>
> *“The paper is hard to follow at many places since only informal theorems are provided throughout the paper”.  *(later)* “[A]ll the technical statements in the main paper are Informal theorems with no rigorous explanations provided”.*
>
> **Response:**
> This is not the case: Rigorous versions of all statements are given in Section 2 (Theorem 2.2, Corollary 2.3, Theorem 2.7, Corollary 2.8, Corollary 2.9 and Corollary 2.11) on pages 7-9. The rationale behind stating informal versions is to make the exposition gentler and first explain the intuition and operational meaning of our results (as also pointed out by Reviewer VnD6). A similar strategy is adopted by many other theory-forward machine learning papers (e.g., Kleinberg et al. (ICML 2018) "An alternative view: When does SGD escape local minima?", Rajeswaran et al. (NeurIPS 2019) "Meta-learning with implicit gradients", Goel et al. (NeurIPS 2019) "Time/accuracy tradeoffs for learning a relu with respect to gaussian marginals", Rajaraman et al. (NeurIPS 2020) "Toward the fundamental limits of imitation learning" ). (We note that a version of Table 1 with explicit constants is provided in Appendix A (Table 2).)
>
> *“Somewhat related to the previous point, some technically deep concepts (such as Wasserstein-1 distance) are introduced in passing and handled in a slightly cavalier manner”.*
>
> **Response:**
> The Wasserstein-$1$ distance was indeed used in Section 1.2 before being formally defined in Section 1.4. Thank you for catching this. We added a reference to Section 1.4.
>
> *“It could have been nice if the authors provided some explanations to the figures -- at the moment, the reader has to guess what the authors intend to say.”*
>
> **Response**
> We added further explanations following Figures 1-2 and Figures 3-4. The additions are typeset in teal blue.
>
> *“The main theorem 2.2 is suboptimal -- optimal network may require width greater than the number of input dimensions n. For image analysis, this corresponds to the number of pixels.”*
>
> **Response:**
> For small $\epsilon$, the complexity estimates in Table 1 imply that the encoder and decoder network's width is indeed *larger* than the input dimension. We realized that the confusion might have arisen from ``flipped'' ordering of the columns in Table 1; in the revised version we swapped the columns to make this clearer. The encoder network's width is lower-bounded by the input dimension $n$ and the decoder's width depends on $\frac{1}{\epsilon}$.
>
> *The constraint examples are too unnatural.*
>
> **Response:**
> In Sections 2.3.1 and 2.3.2 (Corollary 2.9 and Corollary 2.11), we derive constrained universal approximation guarantees for any convex constraint and a broad range of non-convex but geodesically-convex cases.  This covers many examples in convex and non-convex optimization including linear, quadratic, and semidefinite constraints. It also covers instances of geometric deep learning between Riemannian manifolds of interest in computer vision, structured matrix-valued learning, and information theory/information geometry, to name a few. It includes as special cases classes of matrices such as unitary matrices, covariance matrices, or distance matrices; pose manifolds in robotics, functions on spheres, tori spaces, and many other sets and spaces used in modern machine learning. All of these examples arise naturally in a variety of applications from computer vision and robotics to operations research to mathematical finance. We emphasize that our theory also admits arbitrary constraint sets much beyond these familiar examples.
> Your comment made us realize that these facts and connections should be better emphasized; we included a  discussion on page $1$ in the updated manuscript.

---

> ### Author Response · Authors · 2021-11-30
> **it would be great to hear your thoughts**
>
> Dear reviewer w3Aa,
>
> Although the discussion period is all but closed, we would still like to ask whether there are major concerns that we did not address in our response. We thought and worked hard to about how to best respond to your comments and accordingly update the manuscript.
>
> In the light of the above discussions, is your assessment of technical correctness of our work still "1: The main claims of the paper are incorrect or not at all supported by theory or empirical result"? It would be great to hear your thoughts and further input.
>
> Sincerely,
>
> The authors

---

### Official Review · Reviewer_VnD6 · 2021-11-05

**Correctness:** 3
**Technical Novelty And Significance:** 3
**Empirical Novelty And Significance:** 3
**Recommendation:** 8
**Confidence:** 2

**Main Review:**

1. The paper shows the first universal approximation with constraints. The general theory shows that the universal approximation can use the probabilistic transformer to project the outputs in the set $K$. In particular, the probabilistic transformer helps to deal with non-convex constraints in Euclidean space and the geodesic convex constraints in non-Euclidean space, I.e., Riemannian manifold.

2. The geodesically convex constraint seems to be an interesting and promising idea to treat non-convex constraints in the Euclidian space. The assumption on the sectional curvature is uniformly bounded above. Is there any connection that can be said in this case under Ricci curvature tensor?

3. The paper is technically well presented. However, there are still some typos. The authors need to improve their writing.

4. It would be a good idea to provide more examples to demonstrate the general theory.


**Summary Of The Paper:**

The paper under review studies the universal approximation theory with constraints.
For any convex or non-convex compact set, a universal approximation is proved through a probabilistic transformer with constraints. Furthermore, a chart-free universal approximation is established on the Riemannian manifold with geodesically-convex constraints.

**Summary Of The Review:**

The paper seems to address a well-known important class of problems. The universal approximation theorem might make significant impacts in solving optimization problems with constraints.

---

> ### Author Response · Authors · 2021-11-12
> **Ricci curvature might allow to relax some assumptions but it requires serious work**
>
> Thank you for your positive report and suggestions to improve the manuscript. As a general comment, in the updated manuscript we made an earnest effort to hunt down typos and add discussions that improve clarity (also in response to other reviewers' comments). We will keep polishing until the deadline.
>
> *Is there any connection that can be said in this case under Ricci curvature tensor?*
>
> **Response:**
> This is an intriguing thought.  The main property which Assumption 2.10 gave us is strict convexity of the integral functional in Equation 9 (as a function of $\mathbb{P}$) and Lipschitz-continuity of its argmin.  In other words, the "Wasserstein barycenter-map" sending $\mathbb{P}\in \mathcal{P}_1(K)$ to $\overline{\mathbb{P}}\in K$ should be well-defined and Lipschitz.  Though there is little work in the optimal transport of metric geometry literature on the topic, we are aware of the paper by Kim et al. "Wasserstein barycenters over Riemannian manifolds" (2017), specifically Corollary 7.12, which could possibly be used to establish an analogue to Lemma C.3.  Thus, we believe that a generalization of Corollary 2.11 replacing sectional curvature bounds by the less restrictive Ricci curvature bounds is possible. However, it would require some time due to the involved technical subtleties.
>
>
> *It would be a good idea to provide more examples to demonstrate the general theory.*
>
> **Response:**
> We agree that additional examples and illustrations could further improve the paper. We have therefore included a discussion on page $1$ citing examples of manifold constraints arising in robotics, computer vision, and in mathematical finance. We also expanded our numerical experiments to illustrate the variety of geometries addressed by our theory (from convex sets to manifolds to certain ``pathological'' constraint sets). Finally, we promoted the section with numerics from Appendix E to Appendix B.  All these changes are written in teal blue.
>
> We hope that the new discussions, experiments, and illustrations address your comments.

---

### Public Comment · ~Stella_Biderman1 · 2021-11-10
**Does this apply to common models in the real world?**

Your paper shows that there is a parameterized family $\mathcal{F}_n$ that is dense in the space of continuous functions from $\mathbb{R}^n\to K$ as $n\to\infty$.

The parameter for your family however is not something that tends to infinity in practice. In particular, the seminal paper "Scaling Laws for Neural Language Models" shows that one-layer transformers scale exponentially worse than deeper transformers. Pragmatically, there is much more of a limitation on the width of a massive transformer than there is on the depth, and it seems not implausible that ultra large transformers will stop scaling in width and exclusively scale in depth given recent trends in parallelism.

Do you have any thoughts on whether your results hold for a family of neural networks $\mathcal{F}_n$ with constant width and variable depth?

---

> ### Author Response · Authors · 2021-11-12
> **Our results reflect real-world models while opening doors to new possibilities**
>
> We are approximating $f:\mathbb{R}^n\rightarrow K$ by passing through the Wasserstein space $\mathcal{P}_1(K)$ By viewing it as a function in $\mathbb{R}^n \rightarrow \mathcal{P}_1(K)$, this lifting to probability measures allows us to delete all topological obstructions (see Theorem 6 in A. Kratsios and L. Papon’s paper "Universal Approximation Theorems for Differentiable Geometric Deep Learning"). Thus the width of any such model must grow since $\mathcal{P}_1(K)$ is infinite-dimensional (by way of analogy, the universal approximation theorems for *feedforward networks* between Euclidean spaces (such as Park et al. 2021 ICLR paper "Minimum Width for Universal Approximation" or P. Kidger and T. Lyons 2020 NeurIPS paper “Universal Approximation with Deep Narrow Networks”) require the width to grow proportionally to the input and output dimensions; this is the same in our case).  We also note that "unbounded width" occurs in universal approximation theorems for neural models between infinite-dimensional spaces (e.g. Kovachki et al. (2021) “On universal approximation and error bounds for Fourier Neural Operators”).
>
> That said, the rates we derive in Table 1 (and Table 2 in the appendix) show that our approximating transformer’s depth and width both scale as a function of the approximation accuracy.  In particular, its depth scales faster than its width.  This seems to confirm and reflect the experimental evidence, which you refer to, that a transformer’s depth is more expressive than its width.
>
> To address the final question, we do in fact believe that when K is a convex polytope then the approximating transformer can be taken to have fixed width, proportional to the number of K’s vertices, and only its depth scales as a function of the approximation quality.  We did not include this case since the same reasoning does not apply to general convex sets, and, perhaps more interestingly, to general non-convex $K$.

---

### Author Response · Authors · 2021-11-12
**Summary of revisions in round one**

Dear reviewers, dear S. Biderman,

Thank you for the useful feedback and insightful questions. We uploaded a revision of the manuscript that implements many of your suggestions and we are adding the detailed point-by-point responses below. The main changes and additions are:

* New examples of settings and constraint sets in which our theorems hold;
* Updates and expansions of figures and captions to improve clarity;
* Additional numerical experiments to illustrate and verify the theoretical results in Appendix B.3;
* Improvements in notation and missing pointers to definitions of some technical concepts.

 We are looking forward to your further feedback.

Sincerely,

The authors

---

### Author Response · Authors · 2021-11-29
**Do the new discussions address your concerns?**

Dear reviewers,

We want to thank you again for your work. In response to your comments we thought hard about how to improve our explanations and better support the theoretical findings. We feel that this resulted in a better paper. We also made sure to polish our prose and enhance clarity of figures.

Do our new explanations and rewritings address your concerns? There is only a little time left to discuss but if your concerns remain unaddressed we may still try to expand our responses and commit to later edits. It would be great to hear your thoughts.

Sincerely,
The authors

---

### Decision · Program_Chairs · 2022-01-20

**Decision:**

Accept (Spotlight)

**Comment:**

The paper studies an interesting question of whether neural networks can approximate the target function while keep the output in the constraint set. The constraint set is quite natural for  e.g. multi-class classification, where the output has to stay on on the probability manifold. The challenge here is that traditional universal approximation theory only guarantees that $\hat{f}(x) \approx f(x)$, but can not guarantee that $\hat{f}(x)$ lies exactly in the same constraint set as $f(x)$.

The paper made a significant contribution in the theory of deep learning -- It is shown that the neural network can indeed approximate any regular functions while keep the output stay in the regular constraint set. This gives a solid backup in terms of the representation power of neural networks in practice, to represent target functions whose outputs are in certain constraint set (e.g. probabilities).